∂ | **Open Peer Review** | Microbial Genetics | Research Article

# Live tracking of a plant pathogen outbreak reveals rapid and successive, multidecade plasmid reduction

Veronica Roman-Reyna,[1,2] Anuj Sharma,[3] Hannah Toth,[1,2] Zachary Konkel,[1] Nicolle Omiotek,[1,2] Shashanka Murthy,[4] Seth Faith,[2,4] Jason Slot,[1] Francesca Peduto Hand,[1] Erica M. Goss,[3,5] Jonathan M. Jacobs[1,2]

**ABSTRACT**  Quickly understanding the genomic changes that lead to pathogen emergence is necessary to launch mitigation efforts and reduce harm. In this study, we tracked in real time a 2022 bacterial plant disease outbreak in U.S. geraniums (*Pelargonium × hortorum*) caused by Xhp2022, a novel lineage of *Xanthomonas hortorum*. Genomes from 31 Xhp2022 isolates from seven states showed limited chromosomal variation and all contained a single plasmid (p93). Time tree and single nucleotide polymorphism whole-genome analysis estimated that Xhp2022 emerged within the last decade. The phylogenomic analysis determined that p93 resulted from the cointegration of three plasmids (p31, p45, and p66) found sporadically across isolates from previous outbreaks. Although p93 had a 49 kb nucleotide reduction, it retained putative fitness genes, which became predominant in the 2022 outbreak. Overall, we demonstrated, through rapid whole-genome sequencing and analysis, a recent, traceable event of genome reduction for niche adaptation typically observed over millennia in obligate and fastidious pathogens.

**IMPORTANCE**  The geranium industry, valued at $4 million annually, faces an ongoing *Xanthomonas hortorum* pv. pelargonii (Xhp) pathogen outbreak. To track and describe the outbreak, we compared the genome structure across historical and globally distributed isolates. Our research revealed Xhp population has not had chromosome rearrangements since 1974 and has three distinct plasmids. In 2012, we found all three plasmids in individual Xhp isolates. However, in 2022, the three plasmids co-integrated into one plasmid named p93. p93 retained putative fitness genes but lost extraneous genomic material. Our findings show that the 2022 strain group of the bacterial plant pathogen *Xanthomonas hortorum* underwent a plasmid reduction. We also observed several *Xanthomonas* species from different years, hosts, and continents have similar plasmids to p93, possibly due to shared agricultural settings. We noticed parallels between genome efficiency and reduction that we see across millennia with obligate parasites with increased niche specificity.

**KEYWORDS**  *Xanthomonas hortorum*, plasmid, genome structure, WGS

Emerging pathogenic microorganisms threaten human, animal, and plant health. Pathogenic bacteria adapt to changing environments through gene mutations and mobilization. Gene movement occurs within bacterial genomes and between cells. Intracellular DNA movement is usually mediated by insertion sequences (IS), transposases (Tn), and homologous recombination, and among cells, intercellular movement is caused by horizontal gene transfer (HGT). HGT mechanisms are transformation (uptake of extracellular DNA), conjugation (plasmids), and transduction (mediated by bacteriophages).

Address correspondence to Veronica Roman-Reyna, roman-reyna@psu.edu, or Jonathan M. Jacobs, jacobs.1080@osu.edu.

The authors declare no conflict of interest.

See the funding table on p. 11.

10.1128/msystems.00795-23   **1**

Several disease outbreaks and their global dissemination are associated with plasmid-coded genes (1, 2). Plasmids are functional genetic modules that mediate gene transfer, conferring clinically and economically important properties to bacteria (1–5). An example of bacterial adaptation is the global dissemination of antimicrobial resistance genes threatening public health (6). The constant use of antibiotics has led to an increase in hospitals of methicillin-resistant *Staphylococcus aureus* (MRSA). Another example is plant-associated bacteria with copper-resistant genes. Copper is a broad-spectrum biocide used to control foliar pathogens. Frequent application of copper led to the emergence of *Xanthomonas* and *Pseudomonas* copper-resistant isolates (7). Global outbreaks of plant pathogenic bacteria can serve as models to define pathogen emergence mechanisms.

A long-term evolutionary impact of parasitism is genome reduction (8). Over millions of years, hyper niche-specific microorganisms across the symbiotic spectrum experience major genome loss and often are obligate to their host (9). These million-year-long genomic changes are hard to track in real time but play fundamental roles in host adaptation and pathogen specialization (10, 11). Examples of genome reduction and complex genome evolution include bacterial pathogens, such as the 1.1 Mb genome of *Rickettsia* sp. that causes typhus fever, plasmid-driven evolution in *Erwinia tracheiphila*, the *Burkholderia mallei* chromosome size reduction that causes equine disease, and the Xanthomonadaceae xylem-limited plant pathogenic bacteria *Xylella fastidiosa* (2 Mb genome) and *Xanthomonas albilineans* (3 Mb genome) causing Olive Quick decline and leaf scald in sugarcane (10–14).

The environmental horticulture industry, including floriculture crops, is core to the US economy, totaling $4.80 billion. Among these, geranium plants are particularly important, valued at $5.9 million in 2020 (USDA ERS). A destructive disease of geraniums, reported worldwide, is bacterial blight caused by *Xanthomonas hortorum* pv. pelargonii (Xhp) (15). Xhp spreads by plant vegetative- and seed-propagation, irrigation, and wounds (16–18). The reported symptoms in the leaves are water-soaked spots that turn black, wilting leaf margins, or chlorotic V-shaped lesions (Fig. 1A). Stems develop black rot, and the whole plant wilts. Several Xhp outbreaks have been reported with 10%–100% estimated annual losses in greenhouses and in-field conditions (19–22). In 2022, nationwide producers reported a foliar outbreak of geranium bacterial blight caused by *Xanthomonas hortorum* pv. pelargonii (23). Even with recent reports tracking Xhp, little is known about Xhp evolution, functional diversification, and pathogen epidemiology (24).

Advances in genome sequencing applications for genome-resolved epidemiology have allowed immediate tracking of the evolutionary history of human clinical outbreaks but remain widely employed for plant disease epidemics (25, 26). Rapid sequencing and public data dissemination have made the surveillance process an international effort to mitigate economic and food security risks (27). In 2022, there was an outbreak of geranium blight caused by *X. hortorum* pv. pelargonii. We, therefore, used whole genomes from infected leaf tissue to define the nature of the evolution of the geranium outbreak.

## RESULTS

### The 2022 geranium bacterial blight outbreak is due to the emergence of a novel Xhp lineage

We received 31 Xhp isolates from infected geranium collected in 2022 from 18 different U.S. greenhouses across seven states (Table S1). To define the pathogen's identity and evolutionary history of the 2022 nationwide outbreak compared to historical events, we used short-read Illumina sequencing to generate genome assemblies for isolates from the current and historical outbreaks: 31 from 2022 (Xhp2022); three from 1989 (X-1, X-5, and X-7), two from 2011 (OSU776 and OSU777), one from 2012 (OSU778), and one from 2017 (OSU779) (23). For the analysis, we used the Xhp-type strain CFBP 2533 from 1974 as a reference (28). CFBP 2533 was the only Xhp genome available on NCBI when we started the analysis in 2022.

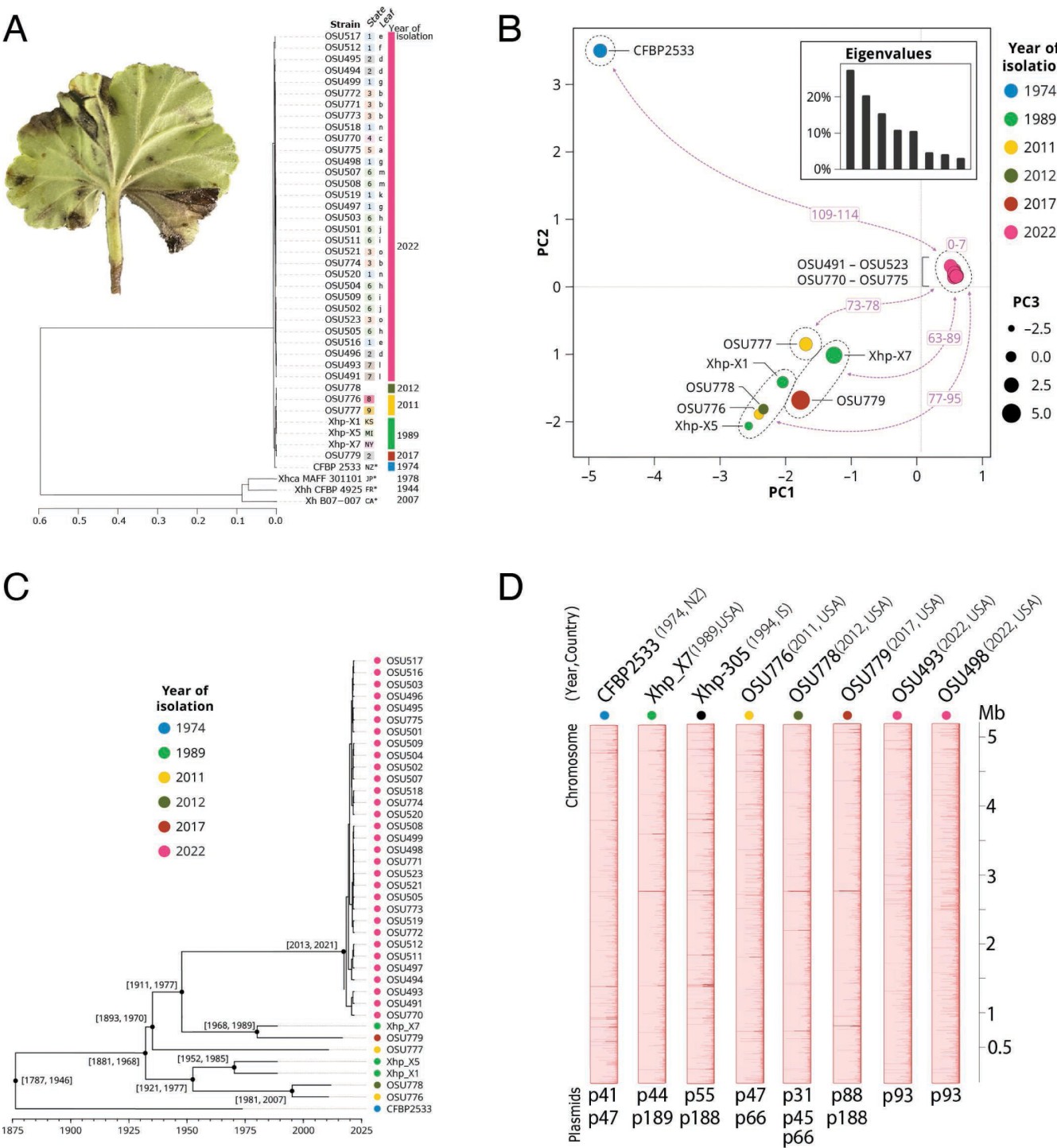

FIG 1 Genomic identification and characterization of the recent geranium Xh2022 bacterial blight outbreak. (A) Symptoms of blight in a geranium leaf and whole-genome average nucleotide identity analysis. Numbers in colored squares indicate geographic location. Lowercase letters show isolates from the same leaf. Colored rectangles indicate the collection year. Xhp, *X. hortorum* pv. pelargonii. Xhh, *X. hortorum* pv. hederae; Xhca, *X. hortorum* pv. carotae. (B) Clustering of the Xhp isolates based on principal component analysis of core chromosomal single nucleotide polymorphisms (SNPs). The number of SNPs differentiating the cluster of 2022 isolates from other clusters is shown in purple. (C) Time-scaled phylogeny of Xhp using an HKY nucleotide substitution model. Brackets indicate a 95% HPD credible interval for the date of each node. Colored dots indicate the collection year. (D) Mauve multiple Xhp chromosome alignment. Each chromosome is rotated to start with the DnaA sequence. Each color shows genetically similar blocks.

To examine genomic distances among the 38 sequenced Xhp isolates, we calculated average nucleotide identity (ANI) among all isolates and included three non-geranium pathogen *X. hortorum* as outgroups. ANI comparisons showed all 38 isolates had more than 95% ANI to other *X. hortorum*, demonstrating that all the same species (Fig. 1A). All 38 isolates had 95.6%–95.9% ANI compared to *X. hortorum* pv. hederae NCPPB939, *X. hortorum* pv. carotae MAFF301101, and *X. hortorum* B07-007 and over 99% ANI (99.8%–99.9%) with the pathovar (pv.) pelargonii-type strain CFBP 2533 (Fig. 1A). This provides additional evidence that they are pv. pelargonii, as they cause disease in geranium. Finally, within the 38 Xhp, Xhp2022 isolates formed a single cluster (99.95% ANI).

We then identified core-genome single nucleotide polymorphism (SNP) for all Xhp isolates (Fig. 1B). The principal component analysis showed five groups based on SNP genotypes. The Xhp2022 isolates formed a distinct group with zero to seven SNPs among them, confirming the ANI comparisons. All other isolates were differentiated from the 2022 outbreak by 63 to 114 SNPs.

To determine when the new Xhp2022 group diverged from other Xhp, we inferred a time-calibrated phylogenetic tree. The analysis revealed that Xhp2022 isolates emerged from a common ancestor between 2013 and 2021 (95% highest posterior density interval; Fig. 1C) with 2017 as the year of highest posterior probability. All the genomics analyses showed that the recent outbreak is due to the emergence of a novel lineage.

## Historical and recent Xhp isolates have similar chromosome content and structure

To further define the genome structure including chromosome and plasmid resolution in Xhp2022 compared to previous outbreaks, we generated Oxford Nanopore long-read complete genomes of seven Xhp isolates from different years: 1989 (X-7), 2011 (OSU776), 2012 (OSU778), 2017 (OSU779), and 2022 (OSU493, OSU498) (Table S1). We used Xhp CFBP 2533 from 1974 as a reference, as the Oxford Nanopore long-read complete genome is publicly available on NCBI (29). We included PacBio long-read sequenced strain Xhp-305 isolated in 1994, as it was uploaded to NCBI in February 2023. Based on the assemblies, there were few differences in whole chromosome synteny across six decades of Xhp isolates. Instead, there was a high variation in plasmid abundance and sizes (Fig. 1D). Based on the assemblies, we compared chromosomes and plasmids separately.

The Xhp chromosomes alignment showed no rearrangements or inversions and one contiguous colinear block (Fig. 1D). The chromosomes contained a total of 4,455 ± 200 genes, with 3,758 being core genes based on the pangenome comparisons (Fig. S1A; Tables S2 and S3). The OSU strains have 100–600 unique genes annotated mainly as hypothetical proteins (Fig. S1A; Table S2 ). The second most abundant genes encoded for genetic information processing are associated with gene movement and host adaptation (Table S3) (30). Therefore, we annotated the chromosomes using the mobile genetic elements (MGE) database mobileOG-db (31). Overall, all seven chromosomes have 202 to 221 MGEs with the same loci (Fig. S1B; Table S4). Notably, these genomes over 60 years did not show significant changes associated with the abundance and location of MGEs (F (7, 32) =0.011, $P = 0.99$).

Virulence factors such as type three secreted effectors are injected directly into plant cells to manipulate immunity and are required for successful host interaction (32). To assess the type three secreted effectors diversity across Xhp chromosomes, we annotated the chromosomes for type three secreted *Xanthomonas* Xop effectors. Chromosomes from all seven long-read genomes have 19 to 20 Xop effectors and share 14 of them in the same loci (Fig. S1C; Table S5).

Next, we assessed codon usage differences across Xhp isolates to determine whether the chromosome is under pressure to increase translation efficiency for bacterial fitness (33). Based on the Dynamic Codon Biaser (DCB) tool, all Xhp chromosomes have a similar codon usage to Xhp CFBP 2533 ($R^2 = 0.999$) (Table S9). Overall, the location and content

of host-associated fitness factors and MGEs indicate the Xhp chromosomes were similar across decades.

The differences in the chromosome were mainly associated with hypothetical proteins that did not affect the chromosome structure. As we cannot explain the emergence of Xhp2022 based on the hypothetical proteins, we focus on the plasmid differences.

## The Xhp 2022 isolates have a reduced plasmid

Plasmids are a primary vector for bacterial HGT and rapid adaptation to environmental change (4). The long-read assemblies revealed that Xhp isolates from previous outbreaks have at least two plasmids, from 31 kb to 189 kb. The Xhp2022 isolates only encoded one plasmid (p93) of 93 kb. Based on plasmid number and size differences among Xhp isolates, we explored the plasmid structure and gene content.

To determine the similarity in p93 plasmid content across Xhp2022 and other *Xanthomonas* species, including historically Xhp sequenced above, we did a BLASTn search to find plasmids with shared identity with p93. The search revealed that Xhp2022

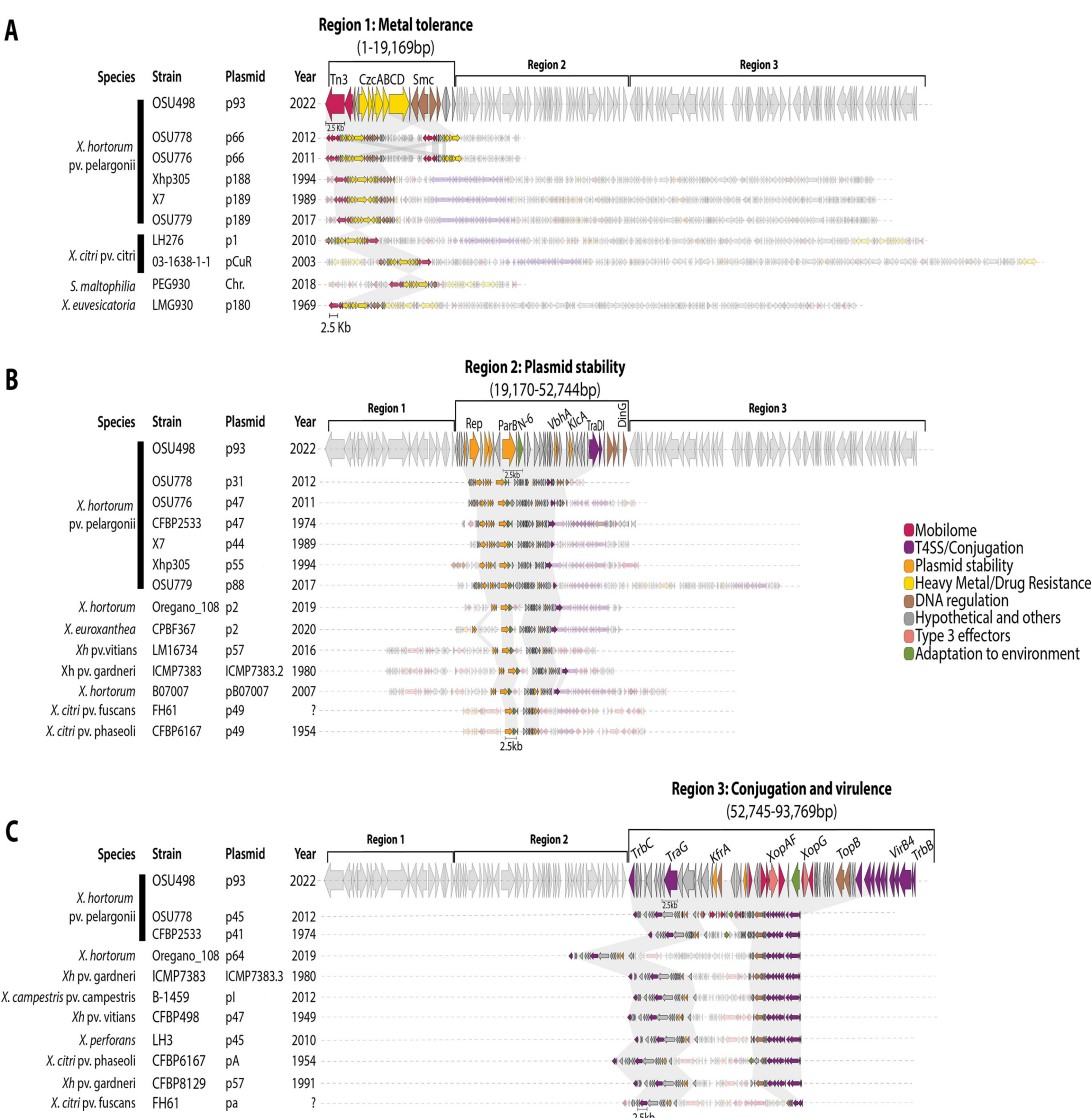

**FIG 2** Gene cluster alignments of Xhp2022 and *Xanthomonas* plasmids. Clinker plasmids protein alignments to the (A) region 1: metal tolerance; (B) region 2: plasmid stability; and (C) region 3: conjugation and virulence strategies of p93. Each arrow represents a coded gene and the color represents a broad classification. The gray translucent connecting squares indicate similarities among plasmids that are above 60%.

p93 contained three distinct regions (regions 1–3) with hits to different plasmids (Table S6). Additional BLASTn using each Xhp 2012 plasmid as a query confirmed the three Xhp2022 p93 regions with 99% identity and with more than 60% coverage (Table S6). A search in the plasmid database PLSDB for p93 and the 2012 plasmids validated the BLASTn results (Table S7) (34). A global synteny alignment showed the p93 regions had similar structure and protein content to their plasmid's hits (greater than 60% alignment sequence similarity) (Fig. 2) (23).

Region 1 had high nucleotide identity (99%) and 20% coverage with Xhp isolates from 1989 (p189), 1994 (p188), 2011 (p66), 2012 (p66), and 2017 (p189) (Table S6). *X. citri* pv. citri and *X. euvesicatoria* plasmids, and *Stenotrophomonas maltophilia* chromosome aligned (92-93% identity) to region 1, which highlights the conservation and mobility of this region outside Xhp. For the protein analysis, region 1 shared proteins with the largest Xhp plasmids from 1989, 1994, 2011, 2012, and 2017 (Fig. 2A). Region 1 contained proteins associated with cobalt/zinc/cadmium efflux system CzcABCD, mobile elements (Tn3 and recombinases), structural maintenance of chromosomes (Smc), and DNA binding (SEC-C and exoribonuclease). All Xhp plasmids shared the same CzcABCD cluster with *X. citri*, *X. euvesicatoria* pLMG930.1, and *S. maltophilia* chromosome.

Region 2 was present in plasmids across Xhp isolates with 99% identity (30% coverage) from 1974 (p47), 1989 (p44), 1994 (p55), 2011 (p47), 2012 (p31), 2017 (p88), and *X. hortorum* Oregano_108 plasmid2, followed by 95−97% identity to other *X. hortorum* pathovars, and *X. citri* pathovars fuscans and phaseoli (Table S6). Region 2 shared these proteins with Xhp plasmids from 1974, 1989, 1994, 2011, 2012, and 2017 (Fig. 2B). Region 2 had genes coding for DNA replication and regulation proteins (Rep, ParB, 3′−5′ exonuclease DinG, and RecD), restriction/modification system (N6 DNA methylase), and orphan antitoxins (VbhA, RelE/ParE, VapC). These proteins were also present in five different *X. hortorum* isolates, *X. euroxanthea,* and two different *X. citri* pathovar plasmids.

Region 3 shared identity (97%−99%, with 50% coverage) with the plasmids from 1974 (p41), 2012 (p45), *X. campestris* pv. *campestris*, other *X. hortorum* pathovars, and *X. citri* pathovars (Table S6). Region 3 shared all proteins with the plasmid p45 from 2012 and partly with p41 from 1974 (Fig. 2C). Region 3 had genes coding for proteins associated with conjugation (Vir and Tra protein cluster), type three secreted effectors (XopAF and XopG), and DNA replication (DNA topoisomerase TopB). In addition, it shares most of the content with other *X. hortorum* pathovars, *X. citri* pv. citri, *X. perforans,* and *X. campestris*. Overall, the p93 regions appear to have conserved functions among historical Xhp plasmids and other *Xanthomonas* species plasmids.

## The Xhp2022 plasmid emerged by plasmid reduction by homologous recombination

Next, we annotated the plasmids using mobileOG-db to assess the plasmid mechanisms for gene movement and stability (31). The three Xhp2022 p93 regions overlapped with p66 and p45 at genes coding for integration/excision proteins (Tn3, *xerC*, recombinase, invertase) and with p31 and p45 at genes encoding DNA repair enzymes (*dinG*, *recBCDF*, exoribonuclease), conjugation machinery proteins (*trb* genes), and with p66 and p31 at genes encoding DNA binding enzymes (*smc*, *sec*-C) (Fig. 3A). The overlapping genes suggested homologous and site-specific recombination as a potential mechanism for plasmid cointegration (Fig. 3B) (2, 35).

To precisely find the expansion events in the Xhp2022 plasmid, we predicted the plasmid mobility, analyzed the phylogenetic relationships of individual proteins in each p93 region, and performed average nucleotide comparisons of whole plasmids. For plasmid mobility, we classified the Xhp plasmids by relaxase type (*mob*) and plasmid transferability (36). We found four relaxases, *mob*F, *mob*P, *mob*H, and *mob*Q. p93 had *mob*F and *mob*P (Table S7). Plasmids similar to region 1 had either *mob*H or *mob*Q. The plasmids that matched region 2 had *mob*F, and region 3 had *mob*P. All plasmids were

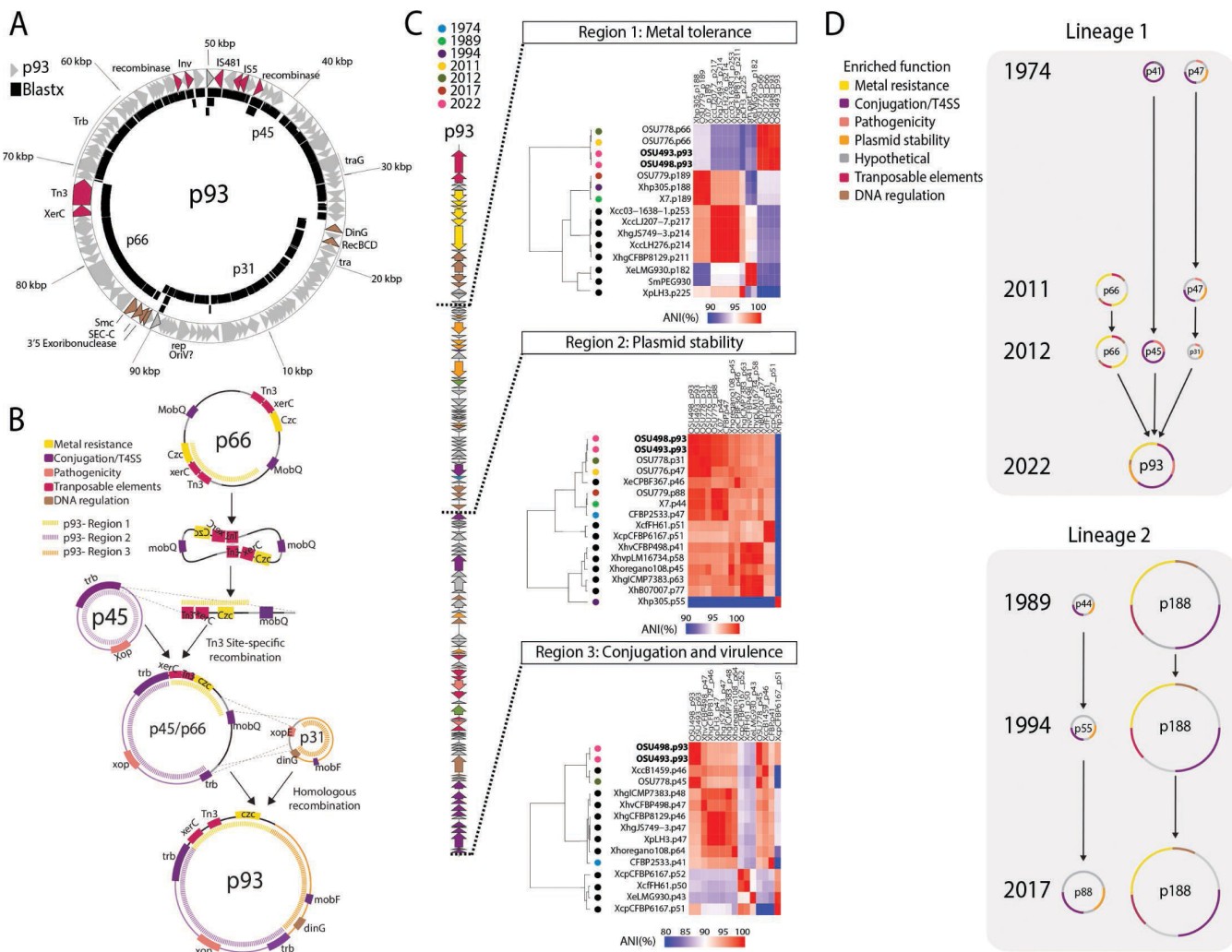

**FIG 3** Evolutionary dynamics of plasmid cointegration in Xhp2022. (A) Overlapping genes among 2022 and 2012 plasmids. The arrows represent p93 genes and the black blocks represent BLASTx (identity > 90%) analysis of each 2012 plasmid against p93. Pink arrows represent transposable elements and brown arrows represent DNA regulation-associated genes. (B) Representation of the cointegration events that led to the p93. First, a Tn3-xerC site-specific recombination event integrated p66 into p45. The identical regions between p31 and p66 allowed a homologous recombination event that integrated p31 into p93. (C) Average nucleotide identity comparisons heatmaps among p93 regions, Xhp plasmids, and other *Xanthomonas* plasmids. The colored circles represent the isolation years for each Xhp isolate. Black circles are other *Xanthomonas* plasmids from NCBI. (D) Xhp plasmid evolution model. Translucent gray squares indicate the two Xhp lineages based on the plasmid structures. Black lines represent the evolution path for each plasmid. Each color in the plasmids indicates enriched function transfers across years.

conjugative or mobilizable except for p44, p47, and p31 from 1989, 2011, and 2012, respectively.

To test the evolutionary origin of each region, we created phylogenetic trees based on representative proteins encoded in each region. From region 1, we selected five genes encoding the Czc cluster and Sec-C. The Czc system is involved in heavy metal resistance (2, 37), and Sec-C is associated with protein secretion (38). The proteins CzcACD clustered with p66 from 2012 and other *X. hortorum* (bootstrap > 99%) (Fig. S2A). CzcB and Sec-C associations were inconclusive due to bootstrap values lower than 90%. The protein clusters suggested that most genes from region 1 come from Xhp but originated from the *Stenotrophomonas* chromosome (bootstrap > 98%) and probably then moved to other *Xanthomonas* species plasmids. For region 2, we selected Rep, ParB, N6 DNA methylase, VbhA, and DinG proteins. Rep, ParB, and N-6, clustered with *X. hortorum*, VbhA clustered with *X. hortorum* and *X. arboricola* (bootstrap >98%) (Fig. S2B). The protein

DinG clustered with several *Xanthomonas* species. The outgroups (bootstrap > 90%) suggested that proteins from region 2 came from Xhp and likely have *Xanthomonas* species plasmids as common donors. For region 3, we selected TadA, TraG, XopAF, XopG, and TopB proteins. All proteins grouped with *X. hortorum* but only TraG had a bootstrap value of 99% (Fig. S2C). The trees suggested that proteins from region 3 originate directly from Xhp.

To estimate relatedness among Xhp plasmids in each region, we calculated ANI values (Fig. 3C). Plasmids associated with region 1 formed two groups, the biggest (>100 kb) plasmids from 1989, 1994, and 2017, and the shortest (66–93 kb) plasmids from 2011, 2012, and 2022. Regions 2 and 3 indicated that Xhp plasmids clustered with other *X. hortorum* isolates. Overall, our results show the origin for each region resulted from the 2012 outbreak with ancestral origins outside Xhp.

Next, we annotated the type three effectors as virulence and transmission traits for the pathogen (32). We found four effectors in Xhp plasmids: TAL effector, *xopAF2*, *xopG1*, and *xopE2* (Table S8). The TAL effector was annotated as a pseudogene and was only present in p47 from 1974. *xopAF2* and *xopG1* were present in p93 and p45 from 2012. *xopE2* was present in 1974, 2011, and 2012 plasmids but not in p93. Based on these results, the only virulence factors present in p93 came from p45 in 2012.

## DISCUSSION

Microorganisms with strong niche adaptations display major genomic reduction (39). Chromosome reduction is an indicator of specialization in obligately host-associated organisms. To our knowledge, bacterial extrachromosomal reduction has not been fully studied in plant pathogens. In this study, we suggest that the predominant Xhp population that caused the bacterial geranium blight outbreak in 2022 had a reduced plasmid. The Xhp2022 plasmid (p93) has elements associated with gene movement and fitness that pointed us to propose that p93 originated from a plasmid cointegration. We propose site-specific and homologous recombination as the events that led to the plasmid reduction (Fig. 3B). Based on the structure of p66, the Tn*3*-xerC integrated within the *trb* operon from p45, as Tn*3* movement does not require similar sequences in the recipient plasmid. The integration in the *trb* operon could explain why *trbC* and *trbB* are no longer syntenic in p93. Since p66 is a dimer plasmid, it is possible that *xerC* resolved the dimer and only a monomer integrated into p45. The integrated p66 fragment has identical regions to p31 that could have led to homologous recombination to integrate p31. The recombination could have excised 20 genes from p66 among *mobQ*, and three genes from p31 (*xopE*, DNA-invertase *hin*, hypothetical protein). We suggest that co-integration into p45 enabled mobilization of p66 and p31, as cointegration is a mechanism for moving non-transferable plasmids from the same host (5, 40–42).

Based on plasmid structure and ANI comparisons of all the Xhp isolates, we propose two Xhp plasmid lineages (Fig. 3D). Xhp isolates from 1974, 2011, 2012, and 2022 belong to lineage 1, and isolates from 1989, 1994, and 2017, are part of lineage 2. The lineage differences are the *trb* and *tra* locus, type three secreted effectors, and two more operons associated with copper and arsenic tolerance (Fig. 2). Likely, after plasmid cointegration, the Xhp population from lineage 1 was enriched, which led to the 2022 outbreak. Plasmid diversification indicates bacterial adaptation to different environmental pressures, as has been described for other plant pathogens like the oncogenic plasmids from *Agrobacterium* (43). Moreover, the distribution of the Xhp2022 lineage across seven states shows the potential for rapid long-distance dissemination of a novel plasmid.

An alternate explanation for the changes in plasmids across decades is genetic drift, particularly associated with population bottlenecks between epidemics. The host cell is responsible for governing the plasmid segregation during cell division, which occurs without selection. The distribution of plasmids into daughter cells could be biased by their location, leading to inadequate segregation (segregational drift). This could have led to some strains having two or three plasmids. Changes in plasmid gene content

during periods when population size is relatively small may result in losses of beneficial mutations from the population, delay in fixation of beneficial mutations, and fixation of neutral mutations or even slightly deleterious mutations linked to beneficial mutations. Over time, this accumulation of changes could lead to the divergence of plasmid lineages or plasmid integration (44).

Plasmid protein phylogenies showed that most genes in p93 are present in *X. hortorum* subgroups but originated from other *Xanthomonas* species or *Stenotrophomonas*. Moreover, plasmid transfers occur among Xanthomonadales species (45, 46). We suggest that several *Xanthomonas* species have similar selection pressures to promote the proliferation of plasmid-containing bacteria and genetic interchange across species (47, 48). That could explain why several plant pathogenic *Xanthomonas* species from different years, hosts, and continents have similar plasmids to p93 (identity > 50%) (Fig. S3). An example of shared selection pressure for Xanthomonadaceae is copper usage for disease management (49). Copper usage promotes the movement of the copper tolerance genes from *X. citri* plasmids and *Stenotrophomonas maltophilia* chromosomes and vice versa (37, 50).

*Xanthomonas* species have similar agricultural settings contributing to pathogen spread, like host and soil microbiome and greenhouses. The microbiome is a source of genetic interchange (51, 52). The soil and host microbiome could be a reservoir for copper tolerance and virulence-related genes horizontally acquired by new isolates (48, 53, 54). Greenhouse facilities might have multiple crops, use vegetative propagation, and irrigation water (splashing water) that leads to pathogens spreading. These agricultural settings shared among plant pathogenic *Xanthomonas* might allow them to interact and share genetic materials.

Overall, we used short and long sequencing to describe the genomic structure of the 2022 Xhp outbreak and compare it to isolates from earlier decades. We provide a comprehensive description of the Xhp outbreak with single nucleotide polymorphisms and whole-genome structure. We propose that an Xhp population with reduced plasmid was predominant in the 2022 outbreak. Based on the minimal differences in phylogenetic distances and SNP variations, we hypothesize that the Xhp2022 nationwide outbreak initial inoculum arrived from a single geranium production location abroad and was then disseminated across the United States, consistent with previous findings (24).

## MATERIALS AND METHODS

### Bacteria isolates and genome sequencing

*X. hortorum* pv. pelargonii isolates were obtained from U.S. nurseries. Bacterial DNA extractions for short-read and long-read sequencing were done with the Monarch Genomic DNA Purification Kit (T3010S, New England Biolabs, Inc.) and Genomic DNA Buffer Set with Genomic-tip 20/G (Cat. No. 19060 and 10223, QIAGEN), respectively. The library for short-read sequences was prepared and sequenced at the Applied Microbiology Services Lab (AMSL)-Ohio State University. Briefly, sequencing libraries were prepared with Illumina DNA prep (Illumina, San Diego, CA), multiplexed with 10 nt dual unique indexes (IDT, Coralville, IA), and sequenced at a target depth of 100× in Illumina NextSeq2000. Long-read sequences were generated with the Oxford Nanopore MinION, Flowcell 9.4, and the Rapid ligation kit (RBK004).

### Short-read analysis

Core chromosomal single nucleotide polymorphisms were identified using the PROK-SNPTREE pipeline (55). Briefly, the quality of the raw reads was verified with FASTQC v0.11.7 (bioinformatics.babraham.ac.uk/projects/fastqc) and the adapters were trimmed with TRIM_GALORE v0.6.5 (56). Clean reads files were aligned to the whole genome assembly of *X. hortorum* pv. pelargonii strain CFBP 2533 (29) using BWA v0.7.17 (57). Variant calling was performed using the HAPLOTYPECALLER tool in GATK V4.1.9.0 to generate base-pair resolved Variant Calling Format (VCF) files (58). For each SNP position,

the nucleotide base for each isolate at that position was determined from base-pair resolved VCF file under stringent filtration parameters (absent if QD <5; reference allele if Alternate allele Depth <10 × Reference allele Depth; and else alternate allele) to generate an allele table. All chromosomal positions with high-quality SNPs were retained. The nucleotides for positions conserved in all isolates were concatenated to generate SNP alignment which was used for subsequent analyses. Principal component analysis was performed in R v4.2.2. A distance matrix was generated from SNP alignment using the r/adegenet package (59) and the principal components were computed using the r/ade4 package (60). The dated phylogenetic tree was generated from the SNP alignment and year of collection in Bayesian Evolutionary Analyses of Sampled Trees (BEAST) v. 2.7.3 (61). Priors included an HKY nucleotide substitution model with empirical base frequencies, an uncorrelated relaxed clock, and a Bayesian Skyline (62). Markov Chain Monte Carlo chain was run for 100 million generations, which produced minimum effective sample sizes of 1,000 for each estimated parameter. From the posterior distribution of 10,001 trees, the maximum clade credibility tree was derived using TREEANNOTATOR v2.4.0 (63) and subsequently visualized in FIGTREE v1.4.4 (tree.bio.ed.ac.uk/software/figtree).

For whole-genome analysis, raw reads were cleaned using Trimmomatic and de-novo assembled using Unicycler and Spades (64–66). Scripts are available at https://github.com/htoth99/Geranium_Project_2022/. The Enveomics website was used for whole-genome Average Nucleotide Identity comparisons and tree generation.

## Long-read analyses

Long raw data were based-called and demultiplexed with Guppy v5 (https://community.nanoporetech.com). Samples were assembled using Flye v. 2.29.2, polished with Homopolish v0.4.1, and annotated using Prokka v1.14.5 (67–69). The chromosomes were rotated to the DnaA using the Python code from Dr. Ralf Koebnik (IRD) and aligned using Progressive Mauve Alignment (70). Functional KEGG categories were assigned using BLAST Koala (71). The effectors for the chromosome and plasmids were predicted using a Xop Xanthomonas proteins database and BLASTx. To determine the mobile genetic elements, we used the MobileOG-db program through the website Proksee (72). To determine codon usage in the chromosomes, we used the website program Dynamic Codon Biaser (http://www.cbdb.info/).

To determine the plasmid relaxase type and their transferability, we used the program MOB-Suite v3.1.2 and the tool MOB-typer (36). To identify similarities to other plasmids, we used the fasta file of each plasmid and used the website database PLSDB with a "Mash dist" search strategy, and 0.1 as the maximum P-value and distance (34). To describe the plasmid similarities with the NCBI nucleotide database, we use the plasmids p93 and the 2012 plasmids as queries in BLASTn. We kept the first 10 hits with >52% coverage and >90% identity. Based on the BLAST results, we selected 5–10 plasmids for protein comparisons. We downloaded the complete accession for each plasmid and annotated with Prokka to have the same annotations. We then perform a whole plasmid protein alignment using the program, Clinker (73). The result analysis with each plasmid was exported as an html and only connections with more than 60% similarity were kept. Colors were edited in Adobe Illustrator.

Based on NCBI and clicker results, we selected five proteins from each plasmid region for further analysis. We used the algorithm BLASTp, the protein as a query, and limited the search for the families Xanthomonadaceae and Pseudomonadaceae. The 100 hits for each gene were downloaded for phylogeny reconstruction. Proteins were aligned and cleaned using MAFFT and ClipKIT (74, 75). We used FastTree to construct trees and identify the root (76). If Pseudomonas species were the root, we chose the next species in line. Based on the FastTree results, we run the analysis using at least 50 sequences with IQ-TREE. Trees were visualized with Figtree. The trees were rooted based on the highest bootstrap value. All trees rooted whole plasmid average nucleotide identity analysis and trees were done with the program Pyani and the ANIb format (77).

Xanthomonadaceae whole-genome tree was built using the website Enveomics and only using complete genomes (chromosome and plasmids) (78). The tree was plotted using the Ward model.

All the pan-chromosome analysis was performed in KBase. First, all genomes were re-annotated with Prokka (v.1.14.15), and then OrthoMCL (v2.0) was used to build the pan-chromosome with default settings. For plotting the results, we used R (v 4.3.2) and the R package UpSetR (version 1.4.0) with default settings.

## ACKNOWLEDGMENTS

We thank the Ohio Supercomputer for bioinformatics support and Dana Martin and Taylor Klass for technical support for *Xanthomonas* isolation and sample processing.

We are grateful for our funding sources, including USDA NIFA FACT-CIN No. 2021-67021-34343 (J.M.J.), Ohio State's Infectious Diseases Institute's Interdisciplinary Seed Grant Program (J.M.J., V.R.-R., and S.F.), the Department of Agriculture's Specialty Crop Block Grant (J.M.J. and F.P.H.), USDA NIFA 2020-67013-31921 (E.M.G.), and USDA Specialty Crop Research Initiative (SCRI) grant 2022-51181-38242.

## AUTHOR AFFILIATIONS

[1]Department of Plant Pathology, The Ohio State University, Columbus, Ohio, USA
[2]Infectious Diseases Institute, The Ohio State University, Columbus, Ohio, USA
[3]Department of Plant Pathology, University of Florida, Gainesville, Florida, USA
[4]Applied Microbiology Services Laboratory, The Ohio State University, Columbus, Ohio, USA
[5]Emerging Pathogens Institute, University of Florida, Gainesville, Florida, USA

## PRESENT ADDRESS

Veronica Roman-Reyna, Department of Plant Pathology and Environmental Microbiology, Pennsylvania State University, University Park, Pennsylvania, USA
Seth Faith, Air Force Research Laboratory, Columbus, Ohio, USA

## AUTHOR ORCIDs

Veronica Roman-Reyna  http://orcid.org/0000-0003-0072-8096
Jason Slot  http://orcid.org/0000-0001-6731-3405
Francesca Peduto Hand  http://orcid.org/0000-0002-5553-0370
Erica M. Goss  http://orcid.org/0000-0003-3512-2107
Jonathan M. Jacobs  http://orcid.org/0000-0002-1553-2013

## FUNDING

| Funder | Grant(s) | Author(s) |
|---|---|---|
| U.S. Department of Agriculture (USDA) | USDA NIFA FACT-CIN No. 2021-67021-34343 | Jonathan M. Jacobs |
| Ohio State University (OSU) | Infectious Diseases Institute's Interdisciplinary Seed Grant Program | Veronica Roman-Reyna |
| | | Veronica Roman-Reyna |
| | | Veronica Roman-Reyna |
| Ohio Department of Agriculture | Specialty Crop Block Grant | Francesca Peduto Hand |
| | | Jonathan M. Jacobs |
| U.S. Department of Agriculture (USDA) | USDA NIFA 2020-67013-31921 | Erica M. Goss |
| U.S. Department of Agriculture (USDA) | USDA Specialty Crop Research Initiative (SCRI) grant 2022-51181-38242 | Erica M. Goss |
| | | Jonathan M. Jacobs |

## AUTHOR CONTRIBUTIONS

Veronica Roman-Reyna, Conceptualization, Investigation, Methodology, Supervision, Visualization, Writing – original draft, Writing – review and editing | Anuj Sharma, Investigation, Methodology, Visualization, Writing – original draft, Writing – review and editing | Hannah Toth, Investigation, Methodology, Visualization | Zachary Konkel, Investigation, Methodology, Writing – review and editing | Nicolle Omiotek, Investigation, Visualization, Writing – review and editing | Shashanka Murthy, Investigation, Methodology, Writing – review and editing | Seth Faith, Investigation, Methodology, Writing – review and editing | Jason Slot, Investigation, Methodology, Supervision, Writing – review and editing | Francesca Peduto Hand, Conceptualization, Supervision | Erica M. Goss, Conceptualization, Investigation, Methodology, Supervision, Visualization, Writing – original draft, Writing – review and editing | Jonathan M. Jacobs, Conceptualization, Investigation, Methodology, Supervision, Visualization, Writing – original draft, Writing – review and editing

## DATA AVAILABILITY

All raw and assembled genomes are deposited in NCBI GenBank and SRA public databases under the project PRJNA846756. The code used in this paper is available at GitHub (https://github.com/htoth99/Geranium_Project_2022/).

## ADDITIONAL FILES

The following material is available online.

### Supplemental Material

**Supplemental figures (mSystems00795-23-s0001.pdf).** Fig. S1 to S3.
**Supplemental tables (mSystems00795-23-s0002.xlsx).** Table S1 to S9.

### Open Peer Review

**PEER REVIEW HISTORY (review-history.pdf).** An accounting of the reviewer comments and feedback.

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
