## [Reviewer comments · mSystems]

Live tracking of a plant pathogen outbreak reveals rapid and successive, multidecade plasmid reduction

Veronica Roman-reyna, Anuj Sharma, Hannah Toth, Zachary Konkel, Nicolle Omiotek, Shashanka Murthy, Seth Faith, Jason Slot, Francesca Peduto Hand, Erica Goss, and Jonathan Jacobs

Corresponding Author(s): Jonathan Jacobs, Ohio State University

Review Timeline:

Submission Date:	July 27, 2023
Editorial Decision:	October 18, 2023
Revision Received:	November 20, 2023
Accepted:	December 15, 2023

Editor: Jon Sanders

Reviewer(s): The reviewers have opted to remain anonymous.

Transaction Report:

DOI: <https://doi.org/10.1128/msystems.00795-23>

October 18, 2023

Prof. Jonathan M Jacobs
Ohio State University
Columbus

Re: mSystems00795-23 (Live tracking of a plant pathogen outbreak reveals rapid and successive, multidecade episome reduction)

Dear Prof. Jonathan M Jacobs:

Thank you for submitting your manuscript to mSystems. We have completed our review and I am pleased to inform you that, in principle, we expect to accept it for publication in mSystems. However, acceptance will not be final until you have adequately addressed the reviewer comments.

Both reviewers found this work to be of substantial interest and generally well-executed.

Still, there were some elements that should be addressed in a minor revision. In particular, as noted by both reviewers, special care should be taken in use of the word 'fitness' to avoid the implication that impacts on selection were actually demonstrated empirically. Reviewer 1 provides some nuanced and specific suggestions on this (and other) points that are worth careful consideration in your revision.

Preparing Revision Guidelines

Please return the manuscript within 60 days; if you cannot complete the modification within this time period, please contact me. If you do not wish to modify the manuscript and prefer to submit it to another journal, please notify me of your decision immediately so that the manuscript may be formally withdrawn from consideration by mSystems.

Sincerely,

Jon Sanders

Editor, mSystems

Journals Department
Reviewer comments:

Reviewer #1 (Comments for the Author):

Review:

- This study is a great fit for the mSystems journal. It is a very thorough study with interesting results about plasmid evolution, inferred from a time series of natural isolates. Most of the results were clearly presented and robust analyses were used.
- The use of PCA in Fig 1C to show genomic relationships was very effective and creative.

The majority of my critique centers around imprecision and inaccuracy in word choice. I organized the points by major critique and minor critique.

Major Critique

- L37 "retained critical fitness genes" is too strong of a claim for an inference-based evolutionary study. Step down the claim by using "putative fitness genes". Similar critique in L48. I suggest that you do a search for "fitness" and verify that you are not describing a hypothesis as if it has already been tested with conclusive results. Similarly, I think you do not have sufficient evidence to say "due to" in L50. That said, if you were inclined to bolster the impact of the study, you could use genetic engineering or plasmid curing to test the hypothesis that the Czc genes confer enhanced fitness to copper.
- L105-106. I suggest that you remove this from the introduction because you never comment on this later in the manuscript. Additionally, the major impact of this paper seems to be the thoroughness of the analysis and the conclusions, not the speed of ... genome sequencing and identification of the pathogen? I believe that it probably took the research team more than two days to define the nature of the evolution of the outbreak lineage.
- L122. "ANI comparisons showed all 38 isolates belonged to the pathovar (pv.) pelargonii" is unclear. What is the ANI threshold required to say the isolates are in the same pathovar as the typestrain? Because Xanthomonas pathovars are not always monophyletic, it might be less confusing to say "All 38 isolates had over xxx% ani with pv. pelargonii type strain CFBP2533, indicating that they belong to pathovar pelargonii." This will also strengthen the statement because I did not find the ANI in the figures or tables.
- L134. I suggest removing "dated to 2017" and just saying the 2013-2021 range. I suspect that you would need more samples / time points to precisely date emergence to a year. This data contradicts the statement from the abstract (L34) that the lineage emerged in the early 2020s. Maybe use 2013-2021 or "emerged within the last decade" in the abstract.
- Is there any chance that you could have overlooked events on the chromosome? The changes in plasmid size are charismatic. But I wonder if anything interesting underlies the "few differences in whole chromosome synteny" (L145). I am not sure how to interpret the Mauve alignments in Fig 1E. I also think that the stacked bar chart in 1F is not easy to interpret. When I look at 1F, I see a dramatic difference in the lengths of the bars. Beyond that, my other main impression is that Kegg is a poor choice for interpreting the functions on the Xanthomonas chromosomes since over half of the genes are "unclassified". I think that Fig 1F should be in the supplemental. For a main-body figure, could you run a pangenome analysis on the chromosome and visualize the results as an UpSet plot? I think that something like this would more clearly demonstrate the "similar chromosome content" result. On that vein, is the statistical test in line 151 meaningful?
- Table S2 does not mention "mobile genetic elements", so I don't understand the logic presented in L151-153. I also don't understand the last sentence of this paragraph. How would genomes have "differences in quantity"?
- I think it might be useful to add a paragraph to the discussion about the role of genetic drift / passive evolutionary forces.
- I recommend rewriting paragraphs L178 and L195 with a different structure. It currently is written chronologically - ANI analysis of Region 1, 2, 3 then synteny analysis of 1, 2,3. It will be more engaging if you write it as "ANI + synteny analysis of 1, ANI +

synteny analysis of 2, ANI + synteny analysis of 3"

- L204 - I don't think you can say that a gene codes for "bacterial fitness", generally. And if you did, I don't think that I would infer this from something annotated as a N6 DNA Methylase. That gene is most likely part of a restriction / modification system. The "antitoxins" are curious. Are they orphan anti-toxins? Or do you infer that they are part of toxin/antitoxin pairs? NCBI CDD search or PaperBlast might let you identify cognate toxins to these anti-toxin genes.

- L232-233 - This sentence is way too speculative to be included in the results. I am not sure that there is necessarily a "driving force" behind this evolutionary change. I don't think that this study disproves a role of genetic drift?

- The font size in many figures is far too small to be readable. Fig 1E. FigS2 -- I recommend that you break this into 3 supplemental figures and increase the size of each panel so that it is readable when printed. Fig 3A -- the annotations are not readable. Remove or increase font sizes. I think 3B would be easier to read with either larger font sizes and/or using colors to indicate regions 1, 2, 3. Fig 3C, the labels above the ANI columns cannot be read.

- Fig 2 -- I think there is a problem with the scale bars. The p93 plasmids seem to be more zoomed in than the plasmids that have a 5 kb marker

- I did not understand "Region 2 and 3 indicated that these plasmids are common for X. hortorum isolates" L 253.

-L324 Which "Qiagen kit"?

- L382. Would you have gotten more diverse hits if you did not limit your blast search to the Xanthomonadaceae family?

Minor Critique on Grammar/word choice:

I suggest using the grammar check on both Google Docs and the "Word Editor" which can be accessed on Microsoft 365.

- co-integration vs. cointegration. Choose a consistent spelling.

- L36 "varying abundance" is non-ideal. Maybe "Found sporadically"

- L39 Did you mean "traceable" instead of tractable?

- Is there any reason to use the term "episome"? I think this just adds more jargon, which can confuse readers.

- You could improve your keywords. For example, I think "plasmid" would be a better keyword than "geranium" for this study.

- You capitalize words that are not necessary to capitalize, e.g. methicillin and "type" in type 3 secretion system.

- Because of the wide margins, when I printed the manuscript, the line numbers were not included.

- CFBP2533 was misspelled at least once. I recommend searching for "2533" and "CFBP" to confirm it is correctly spelled everywhere.

- L125 "no location-related clusters" was confusing.

- L137 "older" than what? Maybe "historical"

- L143 -- "long read sequenced" is imprecise since PacBio and Nanopore assemblies have different types of errors, historically.

- L158 -- use "virulence" instead of pathogenicity

- L187. Move the "Fig 2B" citation to the end of the sentence. With the current location of this reference, I was confused why the results did not mention all of the plasmids shown in the figure.

- L195. I don't think you can use "clinker" as a clear adjective until it becomes much more common as a tool. Say "global synteny" or "gene neighborhood analysis" because those terms are more common / older than the clinker software.

- L 208 you say "T3E" but I think you do not use that abbreviation elsewhere. I suggest that you don't abbreviate this.

- L225 - gene names are not fully italicized in this paragraph

- Add a space between numbers and units, e.g. 5 kb.

- L253 -- Is an adjective missing before "plasmids from"?

- L276 -- XerC should be a protein in this case

- "We suggest that the non-conjugative plasmids p66 and p31 cointegrated into p45 for mobilization" implies that the plasmids made decisions. Rephrase to "co-integration enabled mobilization of p66/p31"

-L294 "higher" than what? rephrase

-L304 "greenhouse practices" is not an agricultural setting, but the greenhouse is.

-L309 remove "for fitness"

-L388 correct "fastree"

Reviewer #2 (Comments for the Author):

The manuscript by Roman-Reyna et al. titled "Live tracking of a plant pathogen outbreak reveal rapid and successive, multidecade episome reduction" demonstrated an example of pathogen evolution through a short period of time. The authors showed that *Xanthomonas hortorum* pv. *pelargonii* (Xhp) underwent genome rearrangement through a period of four decades, with three plasmids integrated into a single plasmid named p93 - which retained critical fitness genes but lost extraneous

genomic material. The analysis of sequencing data is well-structured, and I find that the results of this pathogen live-tracking study would benefit the plant pathology community, as well as the microbiology community at large.

There are several points that I think the authors could address to improve the quality of the manuscript:

1. The authors claimed that the integration of other plasmids into p93 retained critical bacterial fitness genes, but this relies entirely on their codon usage analysis for translation efficiency. A comparison of "fitness" genes on p93 and precursor plasmids would be useful for the authors' claim.
2. The authors speculated that greenhouse settings may facilitate the contact of different *Xanthomonas* lineages and allow them to interact and share genetic material (Line #311-314). I am surprised that the geographical information of where the strains were isolated were not discussed (Fig S3). Was there any overlap in the origins of these strains and their "prospective" relatives?
3. Lineages were proposed based on several genetic differences, especially genes related to copper tolerance. Was there any correlation in copper usage in greenhouses where the strains were isolated and copper usage? Perhaps it is out of scope of this manuscript, but I think an experimental evolution experiment to recreate this genome rearrangement in media amended with copper would be useful to demonstrate the selection pressure that pesticides have on bacterial evolution.

The manuscript by Roman-Reyna et al. titled “Live tracking of a plant pathogen outbreak reveal rapid and successive, multidecade episome reduction” demonstrated an example of pathogen evolution through a short period of time. The authors showed that *Xanthomonas hortorum* pv. *pelargonii* (Xhp) underwent genome rearrangement through a period of four decades, with three plasmids integrated into a single plasmid named p93 – which retained critical fitness genes but lost extraneous genomic material. The analysis of sequencing data is well-structured, and I find that the results of this pathogen live-tracking study would benefit the plant pathology community, as well as the microbiology community at large.

There are several points that I think the authors could address to improve the quality of the manuscript:

1. The authors claimed that the integration of other plasmids into p93 retained **critical bacterial fitness genes**, but this relies entirely on their codon usage analysis for translation efficiency. A comparison of “fitness” genes on p93 and precursor plasmids would be useful for the authors’ claim.
2. The authors speculated that greenhouse settings may facilitate the contact of different *Xanthomonas* lineages and allow them to interact and share genetic material (Line #311-314). I am surprised that the geographical information of where the strains were isolated were not discussed (Fig S3). Was there any overlap in the origins of these strains and their “prospective” relatives?
3. Lineages were proposed based on several genetic differences, especially genes related to copper tolerance. Was there any correlation in copper usage in greenhouses where the strains were isolated and copper usage? Perhaps it is out of scope of this manuscript, but I think an experimental evolution experiment to recreate this genome rearrangement in media amended with copper would be useful to demonstrate the selection pressure that pesticides have on bacterial evolution.

We thank the reviewers for all the valuable comments that helped us improve the manuscript. Please see below the responses to the reviewers.

Response to Reviewer #1

- L37 "retained critical fitness genes" is too strong of a claim for an inference-based evolutionary study. Step down the claim by using "putative fitness genes". Similar critique in L48. I suggest that you do a search for "fitness" and verify that you are not describing a hypothesis as if it has already been tested with conclusive results. Similarly, I think you do not have sufficient evidence to say "due to" in L50. That said, if you were inclined to bolster the impact of the study, you could use genetic engineering or plasmid curing to test the hypothesis that the Czc genes confer enhanced fitness to copper.

We changed the text to: “putative fitness genes”. Since the scope of the paper was about using genomics to describe an emergent pathogen and the potential role of plasmids, we rephrase line 50 to be “Our findings show that the 2022 strain group of the bacterial plant pathogen *Xanthomonas hortorum* underwent a plasmid reduction”.

- L105-106. I suggest that you remove this from the introduction because you never comment on this later in the manuscript. Additionally, the major impact of this paper seems to be the thoroughness of the analysis and the conclusions, not the speed of ... genome sequencing and identification of the pathogen? I believe that it probably took the research team more than two days to define the nature of the evolution of the outbreak lineage.

We removed “quickly (two days)”. We changed it to: “We, therefore, used whole genomes from infected leaf tissue to define the nature of the evolution of the geranium outbreak.”

- L122. "ANI comparisons showed all 38 isolates belonged to the pathovar (pv.) pelargonii" is unclear. What is the ANI threshold required to say the isolates are in the same pathovar as the typestrain? Because Xanthomonas pathovars are not always monophyletic, it might be less confusing to say "All 38 isolates had over xxx% ani with pv. pelargonii type strain CFBP2533, indicating that they belong to pathovar pelargonii." This will also strengthen the statement because I did not find the ANI in the figures or tables.

We rephrase it as: “ANI comparisons showed all 38 isolates had more than 95% ANI to other *X. hortorum*, demonstrating that all the same species (Fig. 1A). All 38 isolates had 95.6 - 95.9% ANI compared to *X. hortorum* pv. hederarum NCPPB939, *X. hortorum* pv. carotae MAFF301101 and *X. hortorum* B07-007 and over 99% ANI (99.8 - 99.9%) with the pathovar (pv.) pelargonii type strain CFBP 2533 (Fig 1A). This provides additional evidence that they are pv. pelargonii, as they cause disease in geranium. Finally, within the 38 Xhp, Xhp2022 isolates formed a single cluster (99.95% ANI).

- L134. I suggest removing "dated to 2017" and just saying the 2013-2021 range. I suspect

that you would need more samples / time points to precisely date emergence to a year. This data contradicts the statement from the abstract (L34) that the lineage emerged in the early 2020s. Maybe use 2013-2021 or "emerged within the last decade" in the abstract.

We changed to “emerged within the last decade”

We changed to: “The analysis revealed that Xhp2022 isolates emerged from a common ancestor between 2013-2021 (95% highest posterior density interval; Fig. 1C) with 2017 as the year of highest posterior probability”.

- Is there any chance that you could have overlooked events on the chromosome? The changes in plasmid size are charasmatic. But I wonder if anything interesting underlies the "few differences in whole chromosome synteny" (L145). I am not sure how to interpret the Mauve alignments in Fig 1E. I also think that the stacked bar chart in 1F is not easy to interpret. When I look at 1F, I see a dramatic difference in the lengths of the bars. Beyond that, my other main impression is that Kegg is a poor choice for interpreting the functions on the Xanthomonas chromosomes since over half of the genes are "unclassified". I think that Fig 1F should be in the supplemental. For a main-body figure, could you run a pangenome analysis on the chromosome and visualize the results as an UpSet plot? I think that something like this would more clearly demonstrate the "similar chromosome content" result. On that vein, is the statistical test in line 151 meaningful?

We did a pangenome analysis of the chromosome and we added the information to the methods section. Since most of the unique genes are hypothetical proteins and the structure of the chromosomes did not change, we added the pangenome (Fig. S1A).

For Fig 1F (stacked bars) we removed the statistical analysis, and we moved the figure to supplemental data as Fig. S1B.

We changed the text to be: “The chromosomes contained a total of 4455_±200 genes, with 3758 being core genes based on the pangenome comparisons (Fig. S1A, Table S2 - S3). The OSU strains have 100-600 unique genes annotated mainly as hypothetical proteins (Fig S1A, Table S2).

We also added: “The differences in the chromosome were mainly associated with hypothetical proteins that did not affect the chromosome structure. As we cannot explain the emergence of Xhp2022 based on the hypothetical proteins, we focus on the plasmid differences”.

In the methods, we added: “All the pan-chromosome analysis was performed in KBase. First, all genomes were re-annotated with Prokka (v.1.14.15), then OrthoMCL (v2.0) was used to build the panchromosome with default settings. For plotting the results, we used R (v 4.3.2) and the R package UpSetR (version 1.4.0) with default settings.”

- Table S2 does not mention "mobile genetic elements", so I don't understand the logic presented in L151-153. I also don't understand the last sentence of this paragraph. How would genomes have "differences in quantity"?

We changed to “The second most abundant genes encoded for Genetic information processing”.

We also added: “Notably, these genomes over a 60-year time span did not show significant changes associated with abundance and location of MGEs”.

- I think it might be useful to add a paragraph to the discussion about the role of genetic drift / passive evolutionary forces.

We added in the discussion: “An alternate explanation for the changes in plasmids across decades is genetic drift, particularly associated with population bottlenecks between epidemics. The host cell is responsible for governing the plasmid segregation during cell division, which occurs without selection. The distribution of plasmids into daughter cells could be biased by their location, leading to inadequate segregation (segregational drift). This could have led to some strains having two or three plasmids. Changes in plasmid gene content during periods when population size is relatively small may result in losses of beneficial mutations from the population, delay in fixation of beneficial mutations, and fixation of neutral mutations or even slightly deleterious mutations linked to beneficial mutations. Over time, this accumulation of changes could lead to divergence of plasmid lineages or leading to plasmid integration (44)”.

- I recommend rewriting paragraphs L178 and L195 with a different structure. It currently is written chronologically - ANI analysis of Region 1, 2, 3 then synteny analysis of 1, 2,3. It will be more engaging if you write it as "ANI + synteny analysis of 1, ANI + synteny analysis of 2, ANI + synteny analysis of 3"

We reorganized the paragraphs:

“To determine the similarity in p93 plasmid content across Xhp2022 and other *Xanthomonas* species, including historically Xhp sequenced above, we did a BLASTn search to determine regions of to find plasmids with shared identity with p93. The BLASTn searches revealed that Xhp2022 p93 contained three distinct regions (Region 1-3) with hits to different plasmids with variable functions (Fig. 2A, Table S5S6). Additional BLASTn using each Xhp 2012 plasmid as a query confirmed the three Xhp2022 p93 regions with 99% identity and with more than 60% coverage (Table S6). A search in the plasmid database PLSDB for p93 and the 2012 plasmids validated the BLASTn results (Table S7) (34). A global synteny alignment showed the p93 regions had similar structure and protein content to their plasmid’s hits (greater than 60% alignment sequence similarity) (Fig. 2) (23).”

“Region 1 had high nucleotide identity (99%) and 20% coverage with Xhp isolates from 1989 (p189), 1994 (p188), 2011 (p66), 2012 (p66), and 2017 (p189) (Fig 2A). *X. citri* pv.

citri and *X. euvesicatoria* plasmids, and *Stenotrophomonas maltophilia* chromosome aligned (92-93% identity) to Region 1, which highlights the conservation and mobility of this region outside Xhp. For the protein analysis, Region 1 shared proteins with the largest Xhp plasmids from 1989, 1994, 2011, 2012, and 2017. Region 1 contained proteins associated with cobalt/zinc/cadmium efflux system CzcABCD, mobile elements (Tn3 and recombinases), structural maintenance of chromosomes (Smc) and DNA binding (SEC-C, and exoribonuclease). All Xhp plasmids shared the same CzcABCD cluster with *X. citri*, *X. euvesicatoria* pLMG930.1, and *S. maltophilia* chromosome (Fig. 2A).”

“Region 2 was present in plasmids across Xhp isolates with 99% identity (30% coverage) from 1974 (p47), 1989 (p44), 1994 (p55), 2011 (p47), 2012 (p31), 2017 (p88), and *X. hortorum* Oregano_108 plasmid2, followed by 95-97% identity to other *X. hortorum* pathovars, and *X. citri* pathovars fuscans and phaseoli (Table S6). Region 2 shared these proteins with Xhp plasmids from 1974, 1989, 1994, 2011, 2012, and 2017 (Fig. 2B). Region 2 had genes coding for DNA replication and regulation proteins (Rep, ParB, 3’-5’ exonuclease DinG, and RecD), restriction/modification system (N6 DNA Methylase), and orphan antitoxins (VbhA, RelE/ParE, VapC). These proteins are also present in five different *X. hortorum* isolates, *X. euroxanthea*, and two different *X. citri* pathovar plasmids.”

“Region 3 shared identity (97-99%, with 50% coverage) with the plasmids from 1974 (p41), 2012 (p45), *X. campestris* pv. *campestris*, other *X. hortorum* pathovars, and *X. citri* pathovars (Fig. 2C). Region 3 shared all proteins with the plasmid p45 from 2012 and partly with p41 from 1974. (Fig. 2C). Region 3 had genes coding for proteins associated with conjugation (Vir and Tra protein cluster), type three secreted effectors (XopAF and XopG), and DNA replication (DNA topoisomerase TopB). Additionally, it shares most of the content with other *X. hortorum* pathovars, *X. citri* pv. *citri*, *X. perforans*, and *X. campestris*. Overall, the p93 regions appear to have conserved functions among historical Xhp plasmids and other *Xanthomonas* species plasmids.”

- L204 - I don't think you can say that a gene codes for "bacterial fitness", generally. And if you did, I don't think that I would infer this from something annotated as a N6 DNA Methylase. That gene is most likely part of a restriction / modification system. The "antitoxins" are curious. Are they orphan anti-toxins? Or do you infer that they are part of toxin/antitoxin pairs? NCBI CDD search or PaperBlast might let you identify cognate toxins to these anti-toxin genes.

We changed “bacterial fitness” to “restriction / modification system.”

I performed a BlastP search using their corresponding toxins but did not find any hits in the plasmids. I then revised the text to include the term 'orphan'.

- L232-233 - This sentence is way too speculative to be included in their results. I am not sure that there is necessarily a "driving force" behind this evolutionary change. I don't think that this study disproves a role of genetic drift?

We removed the sentence from the results.

- The font size in many figures is far too small to be readable.

We increase the font in all figures.

FigS2 -- I recommend that you break this into 3 supplemental figures and increase the size of each panel so that it is readable when printed.

I increased the figure size.

Fig 3A -- the annotations are not readable. Remove or increase font sizes.

I increased the font.

I think 3B would be easier to read with either larger font sizes and/or using colors to indicate regions 1, 2, 3.

Regions were indicated as dotted lines.

Fig 3C, the labels above the ANI columns cannot be read.

I increase the font size.

- Fig 2 -- I think there is a problem with the scale bars. The p93 plasmids seem to be more zoomed in than the plasmids that have a 5 kb marker.

The scale was added for the small and the zoomed-in genes.

- I did not understand "Region 2 and 3 indicated that these plasmids are common for *X. hortorum* isolates" L 253.

Changed to: "Regions 2 and 3 indicated that Xhp plasmids clustered with other *X. hortorum* isolates."

-L324 Which "Qiagen kit"?

Changed to: "Monarch ® Genomic DNA Purification Kit (T3010S, New England Biolabs) NEB kit and Genomic DNA Buffer Set with Genomic-tip 20/G (Cat. No. 19060 and 10223, QIAGEN®)."

- L382. Would you have gotten more diverse hits if you did not limit your blast search to the Xanthomonadaceae family?

I did include Pseudomonadaceae as part of the search.

We added to methods: “We used the algorithm BLASTp, the protein as a query, and limited the search for the Families Xanthomonadaceae and Pseudomonadaceae. The 100 hits for each gene were downloaded for phylogeny reconstruction. Proteins were aligned and cleaned using MAFFT and ClipKIT (74, 75). We used FastTree to construct trees and to identify the root (76). If *Pseudomonas* species were the root, we chose the next species in line.”

I suggest using the grammar check on both Google Docs and the "Word Editor" which can be accessed on Microsoft 365.

We did a grammar check of all the document.

- co-integration vs. cointegration. Choose a consistent spelling.

We changed all the document to be cointegration.

- L36 "varying abundance" is non-ideal. Maybe "Found sporadically"

Changed to found sporadically.

- L39 Did you mean "traceable" instead of tractable?

Yes. Changed to traceable.

- Is there any reason to use the term "episome"? I think this just adds more jargon, which can confuse readers.

Changed to “plasmid”.

- You could improve your keywords. For example, I think "plasmid" would be a better keyword than "geranium" for this study.

Changed to plasmid.

- You capitalize words that are not necessary to capitalize, e.g. methicillin and "type" in type 3 secretion system.

Changed to lowercase.

- Because of the wide margins, when I printed the manuscript, the line numbers were not included.

Margins changed to be 1” on each side.

- CFBP2533 was misspelled at least once. I recommend searching for "2533" and "CFBP" to confirm it is correctly spelled everywhere.

All checked and edited.

- L125 "no location-related clusters" was confusing.

We removed “no location-related clusters” from the text.

- L137 "older" than what? Maybe "historical"

We replaced older with historical.

- L143 -- "long read sequenced" is imprecise since PacBio and Nanopore assemblies have different types of errors, historically.

We added “PacBio long-read sequenced”
We also added “Oxford Nanopore long-read”.

- L158 -- use "virulence" instead of pathogenicity

We changed it to virulence.

- L187. Move the "Fig 2B" citation to the end of the sentence. With the current location of this reference, I was confused why the results did not mention all of the plasmids shown in the figure.

We moved Fig 2B.

- L195. I don't think you can use "clinker" as a clear adjective until it becomes much more common as a tool. Say "global synteny" or "gene neighborhood analysis" because those terms are more common / older than the clinker software.

Changed to synteny.

- L 208 you say "T3E" but i think you do not use that abbreviation elsewhere. I suggest that you don't abbreviate this.

All T3E was changed to “type three secreted effectors”.

- L225 - gene names are not fully italicized in this paragraph

We italicized the genes.

- Add a space between numbers and units, e.g. 5 kb.

We added spaces.

- L253 -- Is an adjective missing before "plasmids from"?

Changed to be “the biggest (>100 Kb) plasmids from 1989, 1994, 2017, and the shortest (66 - 93 Kb) plasmids from 2011, 2012, 2022”

- L276 -- XerC should be a protein in this case

We Changed

- "We suggest that the non-conjugative plasmids p66 and p31 cointegrated into p45 for mobilization" implies that the plasmids made decisions. Rephrase to "co-integration enabled mobilization of p66/p31"

We changed it to: “We suggest that co-integration into p45 enabled mobilization of p66 and p31”

-L294 "higher" than what? Rephrase

Changed to: “plasmid transfers occur among Xanthomonadales species.”

-L304 "greenhouse practices" is not an agricultural setting, but the greenhouse is.

Changed.

-L309 remove "for fitness"

We removed “for fitness”.

-L388 correct "fastree"

We changed to FastTree.

Reviewer #2 (Comments for the Author):

1. The authors claimed that the integration of other plasmids into p93 retained critical bacterial fitness genes, but this relies entirely on their codon usage analysis for translation efficiency. A comparison of "fitness" genes on p93 and precursor plasmids would be useful for the authors' claim.

We changed it to “retained putative fitness genes”.

2. The authors speculated that greenhouse settings may facilitate the contact of different *Xanthomonas* lineages and allow them to interact and share genetic material (Line #311-314). I am surprised that the geographical information of where the strains were isolated were not discussed (Fig S3). Was there any overlap in the origins of these strains and their "prospective" relatives?

In the discussion, we added: “Moreover, the distribution of the Xhp2022 lineage across seven states shows the potential for rapid long-distance dissemination of a novel plasmid”.

We did not discuss the geographical information of where the strains were isolated because US geranium production starts from vegetatively propagated plants abroad usually in the tropics with annual introduction from cuttings. Thus, cuttings from a mother plant are transferred to the US and then multiplied, and the widespread prevalence of this lineage and low diversity suggests a single source entry rather than an origin in the US. Moreover, subpopulations emerging from this outbreak and risk assessment of interstate movement were beyond the scope of this research but highlighted an important question to tackle for future bacterial blight management.

3. Lineages were proposed based on several genetic differences, especially genes related to copper tolerance. Was there any correlation in copper usage in greenhouses where the strains were isolated and copper usage? Perhaps it is out of scope of this manuscript, but I think an experimental evolution experiment to recreate this genome rearrangement in media amended with copper would be useful to demonstrate the selection pressure that pesticides have on bacterial evolution.

For the copper analysis, we based the analysis on genetic information from historical and current strains and previous publications on copper usage for *Xanthomonas* disease management. We agree that future experiments could include artificial evolution experiments, but the scope of this paper was to use genomics-based surveillance to identify the specific genetic traits of the outbreak strains related to Xhp 2022.

Re: mSystems00795-23R1 (Live tracking of a plant pathogen outbreak reveals rapid and successive, multidecade plasmid reduction)

Dear Prof. Jonathan M Jacobs:

Your manuscript has been accepted, and I am forwarding it to the ASM production staff for publication. Your paper will first be checked to make sure all elements meet the technical requirements. ASM staff will contact you if anything needs to be revised before copyediting and production can begin. Otherwise, you will be notified when your proofs are ready to be viewed.

Featured Image Submissions: If you would like to submit a potential Featured Image, please email a file and a short legend to mSystems@asmusa.org. Please note that we can only consider images that (i) the authors created or own and (ii) have not been previously published. By submitting, you agree that the image can be used under the same terms as the published article. File requirements: square dimensions (4" x 4"), 300 dpi resolution, RGB colorspace, TIF file format.

Sincerely,
Jon Sanders
Editor
mSystems